# SeVA: Learning to Ask Discriminative Queries for Fine-Grained Visual Recognition

## Abstract

Fine-grained visual recognition (FGVR) aims to distinguish categories based on subtle, localized cues. Recent methods use vision–language models to ask questions for visual hints, but typically rely on fixed templates that yield static attributions rather than adaptive, informative queries. This limits their ability to reveal discriminative features critical to fine-grained categorization. In this work, we ask a key question: *how can we ask better questions that are context aware, targeted, and dynamically guide visual reasoning?* We propose the Anchored Self-Questioning Vision Agent (SeVA), an iterative reasoning framework that combines a visual–question-answering model with two large language models acting as a Questioner and a Reasoner. Rather than extracting surface-level attributions, SeVA begins with a coarse prediction and then actively interrogates the image by generating discriminative, context-sensitive sub-questions. A Verifier highlights relevant regions, and the Reasoner integrates accumulated evidence to refine the prediction over multiple rounds. To ensure stable and effective interaction between these components, SeVA introduces two complementary types of semantic anchors: (i) explicit anchors from prior category names that guide early attention, and (ii) implicit anchors from previous predictions that provide a language-based gradient for progressive reasoning. Experiments on standard FGVR benchmarks demonstrate the importance of asking good questions, enabling SeVA to outperform state-of-the-art methods.

## 1 Introduction

Fine-grained visual recognition (FGVR) Wei et al. (2021), inspired by the human ability to distinguish subordinate categories, aims to enable machines to recognize subtle differences within visually similar classes. It has important applications in biodiversity monitoring Van Horn et al. (2018; 2021), medical diagnosis Ridzuan et al. (2022), and intelligent agriculture Yang et al. (2020). The core challenge lies in identifying discriminative features that vary only in localized regions, such as subtle differences in breast coloration between the American and European Robin. To capture such nuances, traditional methods often rely on expert-annotated auxiliary data, including attributes Vedaldi et al. (2014) and part-level labels Zhang et al. (2014), which are labor-intensive and costly to obtain. To reduce this dependency, recent approaches like SMILE Du et al. (2023) and PHE Zheng et al. (2024) attempt to leverage partially labeled datasets to classify known categories and cluster unseen ones.

Recent advances in large language models (LLMs), such as Qwen Team (2024) and ChatGPT OpenAI (2022), offer new opportunities by encoding broad domain knowledge that can compensate for missing fine-grained supervision. While LLMs have shown strong reasoning capabilities in language tasks, their potential in visual domains remains underutilized. Multimodal large language models (MLLMs), trained on paired image-text corpora, extend this capability to visual inputs and offer a promising direction for vision-language reasoning. However, querying MLLMs for each test image at inference time incurs high computational and latency costs, limiting their practicality in large-scale deployment.

To mitigate this challenge, FineR Liu et al. (2024b) proposes a hybrid approach. It first leverages LLM knowledge on a small number of unlabeled training images to infer category names, then builds a semantic classifier using a vision-language model for fast inference. As illustrated in Figure 1a, FineR constructs a fixed set of templated questions around the super-category (e.g., bird) to prompt a VQA model to extract part-level attribute descriptions. These descriptions are then used to inform

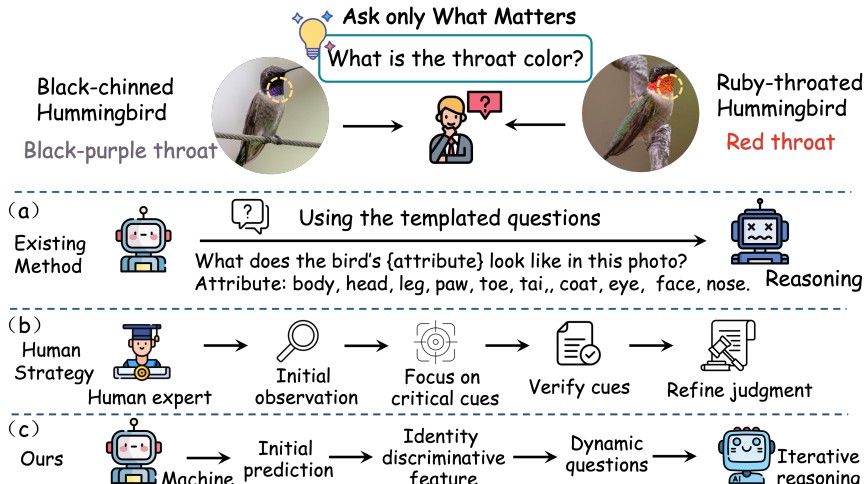

Figure 1: Recognizing visually similar species, such as the Black-chinned and Ruby-throated Hummingbirds, requires identifying subtle, discriminative cues like throat coloration. Existing methods rely on static, templated questions that often overlook these fine details. In contrast, human experts start with an initial hypothesis and iteratively verify key visual regions to refine their judgment. Inspired by this strategy, our method adaptively identifies critical features and formulates targeted questions to guide iterative visual reasoning.

LLM-based category reasoning. While effective in reducing inference costs, this approach has a fundamental limitation: static, generic questions often fail to capture the subtle, category-specific differences that are essential for fine-grained recognition. Moreover, existing methods do not adapt their questioning strategy based on image content, making it difficult to emphasize or revisit the most discriminative cues.

In contrast, human experts approach fine-grained recognition through a dynamic and hypothesis-driven process (Figure 1b). Rather than following a fixed checklist, they use prior knowledge and current visual evidence to propose an initial category, then refine this hypothesis by focusing on key distinguishing features. For example, when identifying whether a hummingbird is Black-chinned or Ruby-throated, an expert does not ask all possible questions about color and shape. Instead, they focus specifically on throat coloration, a defining feature that separates the two. This iterative, selective questioning strategy allows experts to narrow down candidates and refine their judgments efficiently.

Motivated by this cognitive process, we argue that effective FGVR systems should emulate this behavior by generating image-specific, discriminative questions that drive targeted visual reasoning. Static templates are insufficient for this task, as they cannot adapt to the subtle variations that define fine-grained categories. To address this gap, we propose the Anchored Self-Questioning Vision Agent (SeVA) for fine-grained visual recognition that integrates visual querying with LLM-driven analysis through an anchored Questioner–Verifier–Reasoner loop (Figure 1c). In this framework, anchors refer to semantically related categories that act as guiding reference points. These anchors help the system iteratively narrow the candidate space and refine predictions toward the correct fine-grained label.

At the core of this loop, a visual question answering (VQA) model acts as the Verifier, extracting evidence from the image. Two large language models play complementary roles: the Questioner generates discriminative, context-aware sub-questions, and the Reasoner synthesizes the answers into a progressively refined category decision. The process begins with an initial visual query that prompts the VQA model to produce a coarse category prediction. This coarse label serves as an explicit anchor, helping the Questioner identify the most salient visual regions and generate focused, discriminative questions early in the reasoning process. In response to these questions, the Verifier selectively filters out irrelevant visual content and emphasizes features associated with the anchor category, producing targeted sub-answers. These answers are then aggregated by the Reasoner, which tracks and analyzes the evolving evidence across reasoning rounds. As the loop progresses, the system accumulates a trajectory of semantically grounded decisions that form implicit anchors, intermediate predictions that reflect partial understanding of the image. By evaluating their alignment

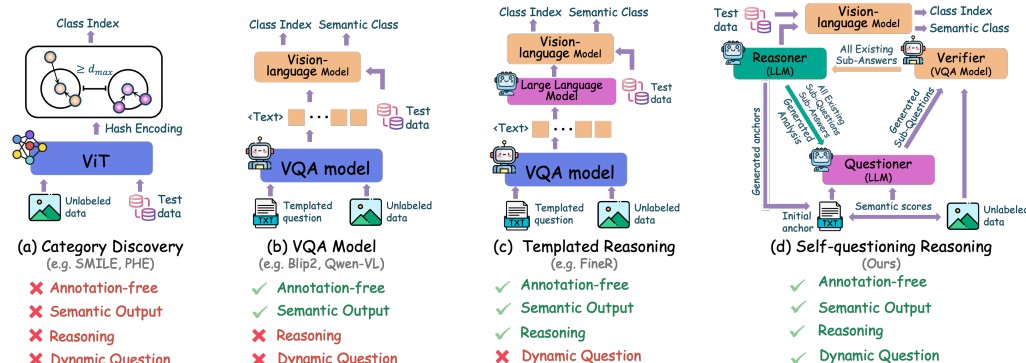

Figure 2: Four paradigms of fine-grained visual recognition. (a) Category discovery requires labeled data and predicts only class indices. (b) VQA models provide semantic insights but lack reasoning capabilities. (c) Templated reasoning generates semantic categories and class indices but often misses critical cues due to rigid question formats. (d) Our method performs iterative, anchor-guided self-questioning, enabling fine-grained recognition without labeled data.

with both visual evidence and linguistic context, these implicit anchors establish a semantic direction in language space Liu et al. (2024c); Yang et al. (2023), guiding the Reasoner toward increasingly accurate and confident predictions.

Our approach combines the strengths of multimodal reasoning, dynamic interaction, and LLM-based world knowledge to go beyond static recognition pipelines. By asking the right questions at the right time, the system can better localize discriminative cues and reason through fine-grained differences, much like a domain expert does. Our main contributions are summarized as follows.

- We propose a self-questioning framework that enables dynamic visual querying and iterative refinement of fine-grained categories through coordinated interaction among a Questioner, Verifier, and Reasoner.
- We introduce semantic anchors, both explicit and implicit, to guide early attention and provide a structured trajectory for progressive reasoning.
- We demonstrate that asking the right questions can achieve superior performance on multiple FGVR benchmarks, resulting in higher accuracy with iterative multimodal queries.

## 2 RELATED WORK

**Fine-grained Visual Recognition.** FGVR Maji et al. (2013); Wah et al. (2011); Wei et al. (2021) aims to distinguish visually similar subordinate categories within a super-category, such as different bird or plant species. Unlike coarse-grained classification, FGVR requires attention to subtle and localized differences. To address this challenge, prior methods have leveraged auxiliary signals such as part annotations Zhang et al. (2014), natural language descriptions He & Peng (2017), web data Krause et al. (2016); Xu et al. (2016); Gebru et al. (2017), and human-in-the-loop supervision Deng et al. (2015); Cui et al. (2016). More recent approaches reduce annotation effort by minimizing reliance on labeled parts Zheng et al. (2017); Ge et al. (2019); Huang & Li (2020). CLEVER Choudhury et al. (2024) replaces expert labels with non-expert descriptions to train a textual similarity model, while FineR Liu et al. (2024b) uses LLMs to reason over visual descriptions without training any dedicated module. *Despite these advances, existing FGVR methods still struggle to extract discriminative cues and fully utilize related categories to support reasoning about novel categories.*

**Category Discovery.** Category discovery aims to group instances of novel categories using prior knowledge from labeled data without class overlap Hsu et al. (2018). Subsequent work improves category separation via rank statistics Han et al. (2021); Zhao & Han (2021), optimal transport Fini et al. (2021), and category relationships Li et al. (2023b); Zhong et al. (2021). Generalized category discovery Vaze et al. (2022); Cao et al. (2022) further allows known and unknown categories to coexist. Recent efforts use contrastive learning Zhao et al. (2023); Pu et al. (2023), self-training Vaze et al. (2024); Chiaroni et al. (2023), and hash-based clustering Du et al. (2023); Zheng et al. (2024) to handle both cases. *However, these methods still rely on labeled data to shape the visual decision space. We instead explore a setting where only a few unlabeled samples are available, and category reasoning is performed through semantic understanding.*

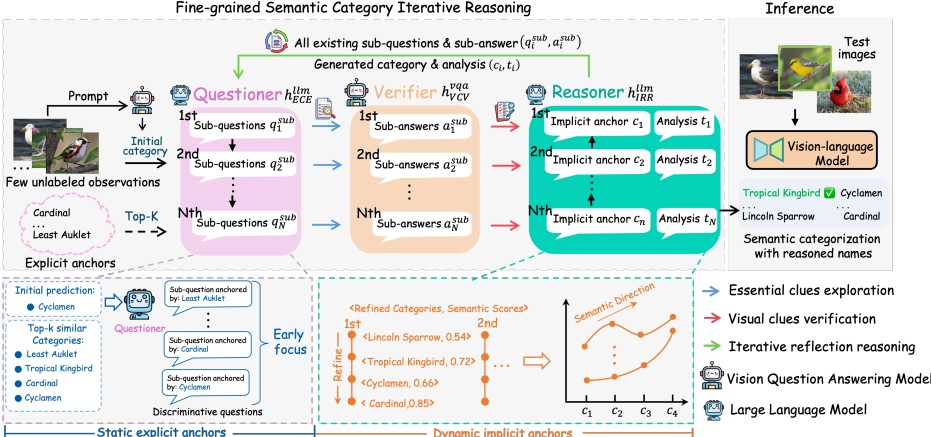

Figure 3: Overview of the proposed SeVA framework. Starting from coarse category predictions as explicit anchors, the Questioner iteratively generates discriminative sub-questions to guide visual attention. The Verifier responds with targeted sub-answers by emphasizing features associated with the anchor category. These answers are then aggregated by the Reasoner, which refines the prediction through multi-round evidence integration. As reasoning progresses, a trajectory of implicit anchors—intermediate semantic states—emerges, providing a directional cue in language space. Finally, the refined category names are used by a vision-language model (VLM) to predict the semantic label for each test image.

**LLM-enhanced Visual Recognition.** LLMs encode rich world knowledge from large-scale text corpora, enabling their use in zero-shot and few-shot vision tasks Tian et al. (2024); Mao et al. (2023); Yan et al. (2023); Menon & Vondrick (2022); Saha et al. (2024). These approaches often generate detailed captions from category names and then classify images using vision–language models Radford et al. (2021). Recent work Conti et al. (2023); Liu et al. (2024b) leverages LLMs or large captioning datasets to support fine-grained classification without fixed label vocabularies. *Yet, most of these methods depend on static templates or external databases and lack the ability to dynamically interact with image content, limiting their capacity to resolve subtle distinctions between fine-grained categories.* The comparison among different paradigms is shown in Fig. 2.

**Chain-of-thought Reasoning.** Chain-of-thought (CoT) prompting Wei et al. (2022) improves LLM reasoning by generating intermediate steps before reaching final answers. Follow-up work Hsieh et al. (2023); Zhu et al. (2023b); Wang et al. (2023) distills CoT reasoning into smaller models for greater consistency. CoT has since been extended to a variety of domains, including multimodal reasoning Zhang et al. (2023); You et al. (2023); Hu et al. (2024), medical diagnosis Liu et al. (2024a); Wu et al. (2025), and visual navigation Zhao et al. (2025); Lin et al. (2025). *In this work, we extend CoT reasoning to FGVR, enabling iterative visual–language reasoning over fine-grained visual cues.*

## 3 ANCHORED SELF-QUESTIONING VISION AGENT

**Problem Formulation.** In this work, we study the problem of fine-grained visual recognition (FGVR) under a new setting where the model is only provided with unlabeled training data $\mathcal{D}^{\text{train}}$ and a set of prior category names $\mathcal{C}^{\text{prior}}$. Different from the previous FGVR task, we do not use any external databases Conti et al. (2023) or auxiliary labeled training data Zheng et al. (2024). Although collecting annotated data for a new domain is often infeasible, a small set of semantically related categories is typically readily available. Thus, our approach further assumes access to a subset of category names, thereby ensuring a realistic application framework.

**Overview.** To recall, our goal is to recognize the fine-grained categories from test image $\mathcal{D}^{\text{test}}$, with only having access to a few unlabeled train data $\mathcal{D}^{\text{train}}$ and their prior category names $\mathcal{C}^{\text{prior}}$. The challenges in such a vision task are how to effectively reason about potential category names from only a few unlabeled observations. To address this, we propose an iterative "Questioner–Verifier-Reasoner" mechanism that employs VQA-based visual understanding and LLM-based analytical reasoning for efficient visual querying. When combined with two complementary semantic anchors, this mechanism enhances collaborative reasoning and precise visual understanding. Each training image triggers the iteration process to bootstrap its category descriptor from the world knowledge encoded

in LLMs, thereby formulating an expandable candidate category pool. This pool is subsequently utilized to on-the-fly predict the semantic label for test image through VLM. An overview of our method is illustrated in Fig. 3.

### 3.1 QUESTIONER

Although the training data comprises only visual images without any expert annotations, it offers rich contextual information that is often overlooked as domain-specific knowledge. To leverage this, we first tap into the world knowledge of VQA model with a prompt template $\rho^{\text{vqa}}$: `What is the species of the {SUPERCLASS} depicted in the provided image?` Here {SUPERCLASS} is the superclass of each FGVR dataset, such as {dog} for Stanford Dog-120 Khosla et al. (2011). This prompt yields a textual response that serves as an initial category prediction for an unlabeled training image. Formally, VQA model $h^{\text{vqa}}(\cdot)$ takes a visual image $x^{\text{train}}$ as input and outputs an object category $c_i^{\text{init}}$:

$$c^{\text{init}} = h^{\text{vqa}}(x^{\text{train}}, \rho^{\text{vqa}}). \tag{1}$$

**Essential cues exploration.** Intuitively, to identify subordinate-level categories, the model is first capable of understanding fine-grained attribute information about the visual object based on the image. However, the abundant information existing in image prevents VQA model from focusing on important visual details, leading to produce similar but incorrect category semantic. To address this, we further employ the strong reasoning capabilities of LLMs to generate meaningful image-related sub-questions, which instruct VQA model to explore whether the image contains essential visual clues about the initial category answer $c^{\text{init}}$. To discover such discriminative cues, we tap into the expert knowledge of LLMs $h_{\text{ECE}}^{\text{llm}}(\cdot)$ with a prompt template $\rho^{\text{llm}}$: `To effectively analyze the {SUPERCLASS} image and determine its species, you should break down the main question into several sub-questions that address the key characteristics of the {SUPERCLASS}.` To ensure sub-question generation toward critical visual cues, LLMs $h_{\text{ECE}}^{\text{llm}}(\cdot)$ are also provided with main question and initial prediction $c^{\text{init}}$ from VQA model. Formally, LLM takes super-category $c^{\text{sup}}$, question $q^{\text{vqa}}$ and answer prediction $c^{\text{init}}$ as input and outputs a list of sub-questions:

$$\boldsymbol{q}^{\text{sub}} = h_{\text{ECE}}^{\text{llm}}(c^{\text{sup}}, \rho^{\text{vqa}}, c^{\text{init}}, \rho^{\text{llm}}), \tag{2}$$

where $\boldsymbol{q}^{\text{sub}} = \{q_1^{\text{sub}}, q_2^{\text{sub}}, \ldots, q_m^{\text{sub}}\}$ denotes the $m$ sub-questions corresponding to the sample $x^{\text{train}}$.

### 3.2 VERIFIER

With the generated informative sub-questions, we introduce a new iterative reasoning mechanism to build our self-questioning vision agent. It consists of two components: (i) Visual cues verification for confirming the presence of visual elements related to the target object in the image; (ii) Iterative reflection reasoning for enhancing the cognitive visual perception capabilities of model with intermediate reasoning chains.

**Visual cues verification.** With generated visual subquestions $\boldsymbol{q}^{\text{sub}}$, we leverage VQA models $h_{\text{VCV}}^{\text{vqa}}(\cdot)$ that excel at identifying visual details (e.g. shape, color) of object by answering image-related questions. For example, if a sub-question is about ear shape, VQA models are prompted to give a description of the ear shape, which is a considerably easier task than directly predicting the subordinate-level category. Concretely, given an image $x^{\text{train}}$ and its corresponding sub-questions $\boldsymbol{q}^{\text{sub}}$, the sub-answers are given as

$$\boldsymbol{a}^{\text{sub}} = h_{\text{VCV}}^{\text{vqa}}(x^{\text{train}}, \boldsymbol{q}^{\text{sub}}), \tag{3}$$

where $\boldsymbol{a}^{\text{sub}} = \{a_1^{\text{sub}}, a_2^{\text{sub}}, \ldots, a_m^{\text{sub}}\}$ denotes sub-answers, representing a set of $m$ visual cues extracted from the image $x^{\text{train}}$.

### 3.3 REASONER

To infer the category name of image $x^{\text{train}}$, we integrate the inherent knowledge and reasoning capabilities of LLMs $h_{\text{Reason}}^{\text{llm}}(\cdot)$ into visual reasoning for perception tasks. This is achieved by

presenting a well-structured prompt, which includes super-category $c^{\text{sup}}$ question prompt $\rho^{\text{vqa}}$, its initial prediction $c^{\text{init}}$, and the associated sub-question–answer pairs $(\boldsymbol{q}^{\text{sub}}, \boldsymbol{a}^{\text{sub}})$. This process can be formally expressed as

$$c = h^{\text{llm}}_{\text{Reason}}(c^{\text{sup}}, \rho^{\text{vqa}}, c^{\text{init}}, \boldsymbol{q}^{\text{sub}}, \boldsymbol{a}^{\text{sub}}), \tag{4}$$

where $c$ is the category name reasoned by LLM.

**Iterative reflection reasoning.** For fine-grained objects, they are hard to be precisely identified with intuitive verification due to subtle differences among subordinate-level categories. This motivates us to incorporate reflection thinking into multi-step decision making, fostering more deliberate reasoning process. To achieve this, we integrate a reasoning component in the form of "analysis" $t_i$, generated before each category $c_i$. These analyses act as a crucial intermediary step, signifying the reflective nature of our deliberative reasoning. As a result, our model iteratively receives visual clues from VQA model and performs reflection reasoning to accomplish the task. This iteration process can be formalized as

$$(c^{\text{sup}}, \rho^{\text{vqa}}, c^{\text{init}}, (\boldsymbol{q}^{\text{sub}}_1, \boldsymbol{a}^{\text{sub}}_1, c_1, t_1), \ldots, (\boldsymbol{q}^{\text{sub}}_N, \boldsymbol{a}^{\text{sub}}_N, c_N, t_N)), \tag{5}$$

where $N$ is the total number of iterations.

The above loop keeps iterating until the agent arrives at a confident category or the number of iterations reaches a predefined maximum. In this way, our agent achieves an explicit reasoning process that enables more nuanced and reflective interactions along the previous reasoning trajectory.

## 3.4 Semantic Anchors

This cooperative, multi-round feedback loop reduces information load per step while progressively uncovering essential cues for fine-grained reasoning. However, it still relies heavily on an initial category hypothesis to formulate sub-questions. Such hypotheses may be insufficiently informative or even unreliable due to the limited fine-grained perceptual capacity of VQA models. Moreover, the Reasoner's lack of direct visual grounding can cause semantic–visual misalignment throughout the iterative process. To mitigate these issues, we design two complementary types of semantic anchors into the iterative reasoning framework.

**Explicit Anchors.** At the beginning of the reasoning loop, we introduce a set of category names as explicit semantic anchors, which guide the Questioner toward generating attribute-specific sub-questions that are highly discriminative. By grounding the sub-question generation on these candidate names, the Verifier's visual attention is steered toward class-specific regions, ensuring that early exploration is both effective and robust.

To identify the semantic anchors $\mathcal{C}^{\text{anchor}}_{\text{top-k}}$ that are similar to the image $x^{\text{train}}$, we retrieve the top-$k$ class names from the prior category set $\mathcal{C}^{\text{prior}}$ in the CLIP-aligned image-text embedding space:

$$\mathcal{C}^{\text{anchor}}_{\text{top-k}} = \arg \max_{\boldsymbol{c} \subseteq \mathcal{C}^{\text{prior}}, |\boldsymbol{c}|=k} \langle \mathcal{E}_{\text{img}}(x), \mathcal{E}_{\text{txt}}(\boldsymbol{c}) \rangle, \tag{6}$$

where $\mathcal{E}_{\text{img}}(\cdot)$ and $\mathcal{E}_{\text{txt}}(\cdot)$ denote the image and text encoders of a vision-language model (e.g., CLIP), and $\langle \cdot, \cdot \rangle$ represents the cosine similarity. This ensures that the selected anchors are not arbitrary but semantically aligned with the image content.

The input image may either belong to one of the retrieved categories or share similar semantics with them. Therefore, we treat them as contextual information (i.e., in-context examples) that guide the large language model (LLM) in generating precise visual sub-questions with the following prompt template $\rho^{\text{llm*}}$: `To effectively analyze the {SUPERCLASS} image and determine its species, you should break down the main question into several sub-questions that can clearly distinguish between the {in-context examples}`. Formally, LLM $h^{\text{llm}}_{\text{ECE}}(\cdot)$ takes super-category $c^{\text{sup}}$, question prompt $\rho^{\text{vqa}}$, initial prediction $c^{\text{init}}$, and in-context examples $\boldsymbol{c}^{\text{anchor}}_{\text{top-}k}$ as input and outputs a list of sub-questions:

$$\boldsymbol{q}^{\text{sub*}} = h^{\text{llm}}_{\text{ECE}}(c^{\text{sup}}, \rho^{\text{vqa}}, c^{\text{init}}, \boldsymbol{c}^{\text{anchor}}_{\text{top-k}}, \rho^{\text{llm*}}), \tag{7}$$

where $\boldsymbol{q}^{\text{sub*}} = \{q^{\text{sub*}}_1, q^{\text{sub*}}_2, \ldots, q^{\text{sub*}}_{m*}\}$ denote the $m$ sub-questions corresponding to the image $x$ and can be viewed as a complement to $\boldsymbol{q}^{\text{sub}}$, ensuring that early exploration remains both informative and robust.

**Implicit Anchors.** As the reasoning loop progresses, the category predictions from previous iterations are aggregated to form implicit anchors. These anchors capture the evolving semantic context and act as a "semantic gradient" Liu et al. (2024c); Du et al. (2024) that refines subsequent reasoning steps. By analyzing the consistency between new sub-answers and previously proposed categories, the Reasoner can filter out spurious predictions and guide the model toward a more accurate final prediction.

To this end, we calculate the similarity between each implicit anchor and vision image as performance metric and incorporate it into LLMs to dynamically refine category prediction within vision-language model. Following Liu et al. (2024c); Du et al. (2024), we envision LLMs as black-box optimizers for vision-language models, which can refine category names based on their performance metrics. Specifically, we maintain a set of implicit anchors $\boldsymbol{c}^{\text{imp}} = (c_1, c_2, \cdots, c_N)$ and their corresponding performance outcomes $\boldsymbol{s}^{\text{imp}} = (s_1, s_2, \cdots, s_N)$. In each iteration, candidate categories are classified as high-performance and low-performance, corresponding to positive and negative categories, respectively. This textual feedback $(\boldsymbol{c}^{\text{imp}}, \boldsymbol{s}^{\text{imp}})$, delivered via in-context prompts, is embedded into LLM prompts, providing an implicit semantic direction Dai et al. (2022).

### 3.5 PREDICTOR

**Expandable category pool.** For each training sample $x \in \mathcal{D}^{\text{train}}$, our iterative reasoning mechanism can reason out its category descriptor based on the world knowledge encoded in LLMs. Consequently, a dynamic and expandable category pool $\mathcal{C}$ can be constructed with the available training data $\mathcal{D}^{\text{train}}$.

**Inference.** Given the reasoned category names in $\mathcal{C}$, we construct a text-based classifier for each category $c \in \mathcal{C}$ using the text-encoder $\mathcal{E}(\cdot)$ of the VLM as:

$$\boldsymbol{w} = \frac{\mathcal{E}_{\text{txt}}(c)}{\|\mathcal{E}_{\text{txt}}(c)\|_2}, \tag{8}$$

where $\|\cdot\|_2$ denotes the two-norm normalization. Using the constructed classifier, test images $x \in \mathcal{D}^{\text{test}}$ can be classified by $\hat{c} = \arg\max_{c \in \mathcal{C}} \langle \mathcal{E}_{\text{img}}(x), \boldsymbol{w} \rangle$, where $\hat{c}$ is the predicted category name with the highest cosine similarity score.

## 4 EXPERIMENT

**Datasets.** We follow previous work Liu et al. (2024b) and adopt five standard datasets for fine-grained visual recognition: Bird-200 Wah et al. (2011), Cars-196 Krause et al. (2013), Dog-120 Khosla et al. (2011), Flower-102 Nilsback & Zisserman (2008), and Pet-37 Parkhi et al. (2012). Consistent with Liu et al. (2024b), we limit the number of unlabeled images per category in $\mathcal{D}^{\text{train}}$ to 3. By default, we assume that only half of the total category names are available as prior knowledge across all methods. Additionally, we conduct experiments under a stricter setting, as proposed in FineR Liu et al. (2024b), where only a few unlabeled training samples are available per category. Due to space limitations, the corresponding results are provided in the supplementary material.

**Evaluation Metrics.** The unconstrained nature of the semantic space makes it difficult to ensure a one-to-one correspondence between ground-truth and predicted categories. Therefore, we adopt two complementary metrics—Clustering Accuracy (cACC) and Semantic Similarity (sACC)—to evaluate performance, following Conti et al. (2023); Liu et al. (2024b). A detailed description of the comparative methods, including their **configurations** and **implementation details**, is provided in the supplementary material.

### 4.1 ABLATION STUDY

As shown in Table 1, we conduct an ablation study to evaluate the effectiveness of individual components within our framework. The baseline model without any additional modules achieves an average cACC of 52.7% and sACC of 68.7%. Self-questioning (SQ) module significantly improves its performance to 58.2% in cACC and 69.1% in sACC, demonstrating its ability to capture task-relevant visual cues. Further, incorporating either the explicit

anchor (EA) or implicit anchor (IA) further boosts performance, with IA providing slightly stronger gains. When both anchors are jointly applied with SQ, the model achieves the best performance (60.6% cACC, 70.9% sACC), demonstrating their complementary effects in facilitating fine-grained reasoning. These results highlight the effectiveness of our full design, where multi-round semantic reasoning and anchor-guided visual cues work synergistically.

Table 1: Ablation study on the effectiveness of different components. "BL" refers to the baseline model without reasoning. "SQ" refers to the self-questioning iteration module, while "EA" and "IA" denote the explicit and implicit anchor mechanisms, respectively.

| Baseline | SQ | EA | IA | Average | |
|---|---|---|---|---|---|
| | | | | cACC | sACC |
| ✓ | ✗ | ✗ | ✗ | 52.7 | 68.7 |
| ✓ | ✓ | ✗ | ✗ | 58.2 | 69.1 |
| ✓ | ✓ | ✓ | ✗ | 59.8 | 69.6 |
| ✓ | ✓ | ✗ | ✓ | 59.8 | 70.7 |
| ✓ | ✓ | ✓ | ✓ | **60.6** | **70.9** |

## 4.2 BENCHMARKING ON FINE-GRAINED DATASETS

We evaluate our method on five widely used fine-grained visual recognition (FGVR) datasets: Bird-200, Cars-196, Dog-120, Flower-102, and Pet-37. Under a setting with only three unlabeled images per class and half of the category names provided as prior knowledge, we report clustering accuracy (cACC) and semantic accuracy (sACC) in Table 2.

Table 2: Comparison of cACC (%) and sACC (%) across five fine-grained benchmarks. All results are obtained using three unlabeled images per class (i.e., $|\mathcal{D}_c^{\text{train}}| = 3$), with half of the class names provided as prior knowledge. "Zero-shot (UB)" and "Finedefics (UB)" denote the upper-bound performance achieved by directly using ground-truth class names for inference and leveraging them for training, respectively. Due to space limitations, we include the results under the same setting as FineR Liu et al. (2019) in the supplementary material to ensure a fair comparison.

| Method | VQA | LLM | Flower-102 | | Dog-120 | | Cars-196 | | Bird-200 | | Pet-37 | | Average | |
|---|---|---|---|---|---|---|---|---|---|---|---|---|---|---|
| | | | cACC | sACC | cACC | sACC | cACC | sACC | cACC | sACC | cACC | sACC | cACC | sACC |
| Zero-shot(UB) | - | - | 69.7 | 77.8 | 56.9 | 75.5 | 63.1 | 66.3 | 57.4 | 80.5 | 81.7 | 87.8 | 65.8 | 77.6 |
| Finedefics (UB) | - | Mistral-7B | 73.2 | 76.4 | 54.8 | 72.2 | 60.8 | 66.2 | 57.9 | 80.6 | 82.3 | 88.3 | 65.8 | 76.7 |
| SMILE | - | - | 42.5 | - | 48.4 | - | 26.2 | - | 32.2 | - | 41.2 | - | 38.1 | - |
| PHE | - | - | 49.0 | - | 43.4 | - | 31.3 | - | 36.4 | - | 48.3 | - | 41.7 | - |
| FineR | BLIP-2 | Qwen-L-7B | 67.4 | 58.2 | 48.5 | 64.8 | 40.0 | 61.3 | 53.4 | 74.2 | 71.7 | 78.9 | 56.2 | 67.5 |
| FineR | BLIP-2 | ChatGPT | 63.0 | 55.8 | 52.9 | 68.1 | 52.0 | 63.4 | 54.8 | 74.6 | 75.6 | 80.0 | 59.7 | 68.4 |
| MiniGPT4 | MiniGPT4 | - | 59.5 | 54.6 | 44.2 | 61.7 | 50.3 | 61.0 | 40.9 | 67.4 | 72.8 | 72.5 | 53.5 | 63.4 |
| MiniGPT4+Ours | MiniGPT4 | Qwen-L-7B | 62.5 | 57.9 | 46.4 | 63.2 | 51.1 | 61.6 | 43.3 | 68.3 | 77.2 | 75.7 | 56.1 | 65.3 |
| Qwen-VL | Qwen-VL | - | 76.7 | 63.9 | 44.6 | 53.1 | 51.1 | 54.6 | 52.8 | 63.4 | 86.0 | 86.2 | 62.2 | 64.2 |
| Qwen-VL+Ours | Qwen-VL | Qwen-L-7B | 81.9 | 64.2 | 76.7 | 63.9 | 51.3 | 55.9 | 55.4 | 69.5 | 86.3 | 86.4 | 70.3 | 68 |
| BLIP-2 | BLIP-2 | - | 68.0 | 66.7 | 45.4 | 65.2 | 53.1 | 62.8 | 44.7 | 71.6 | 73.1 | 78.3 | 56.9 | 69.0 |
| BLIP-2+Ours | BLIP-2 | Qwen-L-7B | 71.9 | **67.1** | 50.3 | **68.5** | **53.2** | **64** | 49.6 | **73.8** | 78.0 | 81.4 | 60.6 | **71.0** |

Compared to traditional category discovery methods like SMILE Du et al. (2023) and PHE Zheng et al. (2024) that require labeled data, our method achieves significantly better performance by leveraging the rich expert knowledge from LLMs. When using the same VQA model (Blip2) and LLM (Qwen-L-7B), our approach outperforms FineR by large margins (e.g., +8.9% cACC on Bird-200), surpassing even FineR variants that use stronger LLMs such as ChatGPT. We further integrate different VQA models (e.g., MiniGPT-4, QwenVL, and BLIP-2) into our framework and observe consistent performance improvements across all architectures, demonstrating its strong generalization capability. While zero-shot Radford et al. (2021) and Finedefics He et al. (2025) directly leverage all ground-truth category names for inference or training, our method remains competitive with or even surpasses these upper bounds on several benchmarks. For instance, on the Flower-102 and Pet-37 datasets, our method achieves cACC scores of 81.9% and 86.3%, respectively, exceeding the corresponding results of Finedefics (73.2% and 82.3%).

**Qualitative comparison.** We visualize and compare the predictions of different methods in Fig. 6 on the Dog-120 and Flower-102 datasets. On the Dog-120 dataset (top row), our method correctly identifies the specific dog species as "African Hunting Dog," accurately capturing the nuanced visual feature of irregularly patterned fur, which distinguishes it from

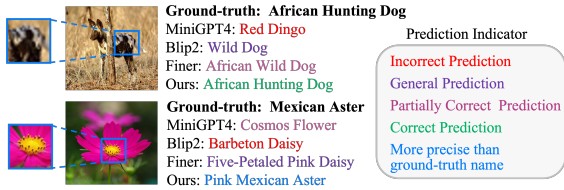

Figure 4: Qualitative comparisons on the Dog-120 and Flower-102 datasets.

visually similar breeds such as the "Red Dingo." In contrast, BLIP-2 tends to predict a coarse-grained category of "Wild Dog," as it does not account for the fine-grained details crucial to FGVR. Although

FineR outputs the category name "African Wild Dog," which is semantically close to "African Hunting Dog," our method produces the biologically accurate label. A similar trend can be observed in the case of "Mexican Aster." Notably, in the case, our method even outperforms the ground-truth by classifying the flower more precisely as "Pink Mexican Aster," guided by the attribute cue "primary flower color: pink" during reasoning.

## 4.3 FURTHER ANALYSIS

**Dynamic self-questioning enabling focused discriminative reasoning.** In Fig. 5, we present case study to illustrate the proposed self-questioning reasoning process. As illustrated in Fig. 5, the model starts from a coarse initial prediction (i.e., Seagull) and progressively refines its hypothesis by asking attribute-specific questions, mimicking human-like elimination and verification. Specifically, the model first hypothesizes Ring-billed Gull based on a yellow bill with a black ring, but subsequently rejects it upon identifying pink legs. It then considers and rules out Glaucous Gull and Western Gull based on bill and back coloration, ultimately converging on the correct fine-grained category Herring Gull when all discriminative features align. This demonstrates that our model can dynamically incorporates visual cues to filter semantically similar distractors.

[Initial category prediction: Seagull]
1. Q: Does the bird have a yellow bill with a black ring?
   → A: ✅ Yes → Accepted ✅.
   → Hypothesis: Ring-billed Gull. Let's verify key attributes.
2. Q: Are the legs yellow?
   → A: ❌ No (they are pink) → Rejected
   Hypothesis (updated): Glaucous Gull.
3. Q: Is there a red spot on the bill?
   → A: ❌ No → Rejected
   → Hypothesis (updated) : Western Gull
4. Q: Is the back dark grey?
   → A: ❌ No → Rejected
   → Hypothesis (updated) : Herring Gull
5. Q: Are the legs pink and the back light grey?
   → A: ✅ Yes → Accepted ✅

Herring Gull

Figure 5: The illustration of our self-questioning reasoning process on the Bird-200 dataset. Additional reasoning trajectories are provided in the supplementary material.

**Progressive refinement of semantic confidence via implicit anchors.** To examine the effectiveness of implicit anchors, we visualize the semantic confidence trajectories over multiple reasoning rounds. As shown in Tab. 3, the IA mechanism not only dynamically adjusts the reasoning direction but also facilitates the model's self-calibration from noisy initial predictions, ultimately evolving to high-confidence decisions. At the early stages of inference, the model produces low-confidence predictions due to insufficient information. However, with the incorporation of IA, semantic knowledge is incrementally accumulated across rounds, forming semantic direction that steers the reasoning process toward the correct category. This behavior also highlights the robustness and progressive refinement capability of our method.

Table 3: The prediction trajectories of category names and corresponding image-text similarity scores across reasoning rounds on the Pet-37 dataset. Green text indicates correct predictions; red text indicates incorrect ones. "0" indicates the initial category prediction.

| Round | 0 | 1 | 2 | 3 | 4 | 5 |
|---|---|---|---|---|---|---|
| Ours w/o IA | Dog 0.26 | Pomeranian 0.29 | Pomeranian 0.29 | Spitz 0.29 | Spitz 0.29 | Spitz 0.29 |
| Ours | Dog 0.26 | Pug 0.21 | Pomeranian 0.29 | Samoyed **0.31** | Pomeranian 0.29 | Samoyed **0.31** |
| Ours w/o IA | Coonhound 0.19 | Persian Cat 0.33 | Scottish Fold 0.27 | Persian 0.32 | British Shorthair 27.08 | Persian 0.32 |
| Ours | Coonhound 0.19 | Maine Coon **0.33** | Scottish Fold 0.27 | Maine Coon **0.33** | Maine Coon **0.33** | Maine Coon **0.33** |

**Sensitivity analysis.** Due to space constraints, we present the detailed sensitivity analysis in the supplementary material. This includes a detailed study of the impact of key hyperparameters (e.g., the number of anchors $k$, the number of reasoning rounds $N$) as well as the effect of model size on final performance.

## 5 CONCLUSION

In this work, we present SeVA, a novel self-questioning vision agent that performs fine-grained visual recognition through a dynamic, iterative reasoning process. Inspired by how human experts progressively refine their hypotheses, SeVA integrates multimodal querying with LLM-driven reasoning in a cooperative Questioner–Verifier–Reasoner framework. By introducing explicit and implicit semantic anchors, our method adaptively localizes discriminative visual cues and progressively narrows the category space, enabling robust recognition. Experiment results show that our training-free framework surpasses state-of-the-art methods across several FGVR benchmarks.

## 6 ETHICS STATEMENT

This work does not involve human subjects, sensitive personal data, or applications that may cause societal harm.

## 7 REPRODUCIBILITY STATEMENT

All datasets used in this work are publicly available and are described in detail in the Experiments section of the main paper. The implementation details, model architectures, training procedures, and hyperparameters are provided in the Appendix. All source code and scripts to reproduce our experiments will be made publicly available upon acceptance.

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

# 8 LLM USAGE STATEMENT

We used large language models (LLMs) only for minor editing purposes, such as improving grammar and writing clarity. They were not used for research ideation, experiment design, analysis, or generating scientific content.

# 9 APPENDIX

## 9.1 PRELIMINARIES

**Vision-language models (VLMs)**, such as CLIP Radford et al. (2021), are designed to produce visual representations via natural language supervision in a contrastive learning setting. It uses 400 million image-text pairs to train the visual and text encoders, where image features from an image encoder $\mathcal{E}_{\text{img}}(\cdot)$ and text features from a text encoder $\mathcal{E}_{\text{txt}}(\cdot)$ are aligned within a unified embedding space. VLMs have demonstrated excellent zero-shot transfer performance on unseen datasets, when provided the subset of category names as prior. For zero-shot inference, a VLM $h^{\text{vlm}}(\cdot)$ can classify a query image $x$ into possible categories $\mathcal{C}$ based on the maximum cosine similarity, denoted as $\hat{c} = \arg\max_{c \in \mathcal{C}} \langle \mathcal{E}_{\text{img}}(x), \mathcal{E}_{\text{txt}}(c) \rangle$.

**Large language models (LLMs)**, such as Qwen Team (2024) and ChatGPT OpenAI (2022) are trained on large internet-scale text corpora, encoding rich world knowledge within their weight parameters. This knowledge inherently encompasses a wide array of descriptions regarding various object concepts, enabling LLMs to serve as knowledge containers for text generation and comprehension in open-ended scenarios. Some researchers typically steer LLMs to understand the task description and generate accurate output by incorporating a few examples within the prompt—a paradigm known as in-context learning (ICL). Further advancements extend ICL to zero-shot reasoning by carefully crafting prompts to provide clear instructions and context to the model. Formally, a LLM, denoted as $h^{\text{llm}} : (\rho^{\text{llm}}) \rightarrow T^{\text{llm}}$, takes context prompt $\rho^{\text{llm}}$ as input, and maps them to a text output sequence $T^{\text{llm}}$.

**Visual question answering (VQA) models**, such as BLIP-2 Li et al. (2023a), MiniGPT-4 Zhu et al. (2023a), integrate the pre-trained LLM with a vision encoder through a projection head, enabling the entire system to be jointly trained in an end-to-end manner. This architecture allows VQA models to effectively process visual images and directly generate final answers to questions, showcasing strong interactive capabilities as intelligent assistants. During inference, a VQA model, denoted as $h^{\text{vqa}} : (x, \rho^{\text{vqa}}) \rightarrow T^{\text{vqa}}$, receives an image $x$ and a textual question prompt $\rho^{\text{vqa}}$ as inputs, and outputs a textual answer $T^{\text{vqa}}$.

**Remark.** VLMs can effectively perform vision recognition tasks when the category names are known. However, in practical applications, acquiring all categories from test set is neither realistic nor feasible. A straightforward approach is to utilize existing VQA models to conduct visual reasoning on images and reason over visual input using question templates. Nevertheless, this direct-response approach often results in errors, especially when the visual task requires expert-level knowledge and fine-grained analytical reasoning. Some works Wei et al. (2022); Yao et al. (2023) demonstrates that LLMs can benefit from generating intermediate natural language rationales that guide the final answer. This motivates us to propose an iterative reasoning framework that combines a visual–question-answering model with two large language models acting as a Questioner and a Reasoner, enabling fine-grained category inference with cognitive visual reasoning capabilities.

## 9.2 IMPLEMENTATION DETAILS

We employ Qwen-L-7B as both the Questioner and the Reasoner. For the Verifier, we adopt three pre-trained vision-language models (VLMs)—BLIP-2 Li et al. (2023a), MiniGPT-4 Zhu et al. (2023a), and LLaVA Liu et al. (2023)—as visual question answering (VQA) modules to enable comprehensive comparison. It is important to emphasize that all VQA models employed in our experiment are pre-trained only and are not fine-tuned on any downstream datasets to guarantee zero-shot generalizability. By default, we utilize a frozen CLIP ViT-B/16 model Radford et al. (2021) as the visual and text encoder to process the image-text data and extract the corresponding embeddings. The

Prediction Indicator: Correct Prediction, Partially Correct Prediction, General Prediction, Incorrect Prediction, More precise than ground-truth name

**Ground-truth: African Hunting Dog**
MiniGPT4: Red Dingo
Blip2: Wild Dog
FineR: African Wild Dog
Ours: African Hunting Dog

**Ground-truth: Mexican Aster**
MiniGPT4: Cosmos Flower
Blip2: Barbeton Daisy
FineR: Five-Petaled Pink Daisy
Ours: Pink Mexican Aster

**Ground-truth: Maine Coon**
MiniGPT4: Long-Haired Domestic Cat
Blip2: Long Haired Cat
FineR: British Longhair
Ours: Maine Coon

**Ground-truth: Sealyham Terrier**
MiniGPT4: Wire Fox Terrier
Blip2: Wire-Haired Fox Terrier
FineR: Wire-Haired Fox Terrier
Ours: Sealyham Terrier

**Ground-truth: Primula**
MiniGPT4: Pink And Yellow Flower
Blip2: Nasturtium
FineR: Purple And Yellow Viola
Ours: Primula

**Ground-truth: Miniature Pinscher**
MiniGPT4: Primroses
Blip2: Scottish Terrier
FineR: Manchester Terrier
Ours: Miniature Pinscher

Dog-120    Flower-102    Pet-37

Figure 6: Qualitative comparisons on Dog-120, Flower-102, and Pet-37 datasets.

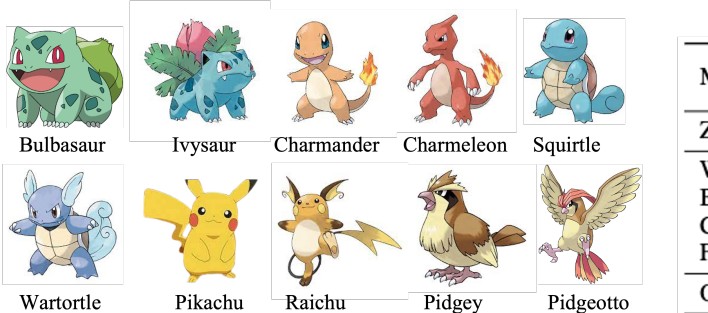

Bulbasaur    Ivysaur    Charmander    Charmeleon    Squirtle

Wartortle    Pikachu    Raichu    Pidgey    Pidgeotto

(a) Ten Pokemon categories and their names

| Method | Pokemon-10 | |
| --- | --- | --- |
| | cACC | sACC |
| Zero-shot (UB) | 70.8 | 89.2 |
| WordNet | 34.6 | 33.1 |
| BLIP-2 | 32.3 | 55.4 |
| CaSED | 39.2 | 55.7 |
| FineR | 70.8 | 81.6 |
| Ours | **71.6** | **83.2** |

(b) Comparison on pokemon dataset

Figure 7: Comparison on the novel Pokemon dataset, with 3 images per category for discovery and 10 for evaluation.

hyperparameters are set as follows: the top-$k$ explicit anchors are fixed at $k = 5$, and the number of reasoning iterations is also set to 5 unless otherwise specified.

Table 4: Quantitative comparison on fine-grained benchmarks under low-resource settings. Following FineR Liu et al. (2024b), we adopt a strict setup where only a few unlabeled training samples are available per category (i.e., $\mathcal{D}_c^{\text{train}} = 3$).

| Method | VQA | LLM | Flower-102 | | Dog-120 | | Car-196 | | Bird-200 | | Pet-37 | | Average | |
| --- | --- | --- | --- | --- | --- | --- | --- | --- | --- | --- | --- | --- | --- | --- |
| | | | cACC | sACC | cACC | sACC | cACC | sACC | cACC | sACC | cACC | sACC | cACC | sACC |
| Zero-shot (UB) | - | - | 69.7 | 77.8 | 56.9 | 75.5 | 63.1 | 66.3 | 57.4 | 80.5 | 81.7 | 87.8 | 65.8 | 77.6 |
| CLIP-Sinkhorn Caron et al. (2020) | - | - | 30.9 | - | 12.6 | - | 18.1 | - | 23.5 | - | 23.1 | - | 21.6 | - |
| DINO-Sinkhorn Caron et al. (2020) | - | - | 17.9 | - | 11.2 | - | 7.4 | - | 13.5 | - | 5.2 | - | 19.1 | - |
| Kmeans Ahmed et al. (2020) | - | - | 66.9 | - | 16.4 | - | 30.6 | - | 36.6 | - | 32.8 | - | 36.7 | - |
| WordNet Miller (1995) | - | - | 42.1 | 49.8 | 53.9 | 49.8 | 18.3 | 33.3 | 39.3 | 57.7 | 55.4 | 61.9 | 41.8 | 54.7 |
| BLIP-2 Li et al. (2023a) | BLIP-2 | - | 61.9 | 59.1 | 39 | 59.1 | 43.1 | 57.9 | 30.9 | 56.8 | 61.3 | 60.5 | 47.2 | 58.6 |
| CLEVER Choudhury et al. (2024) | - | - | 6.2 | - | - | - | - | - | 7.9 | - | - | - | - | - |
| SCD Han et al. (2023) | - | - | - | - | 57.9 | - | - | - | 46.5 | - | - | - | - | - |
| CaSED Conti et al. (2023) | - | - | 67.2 | 52.3 | 38.0 | 52.3 | 26.9 | 41.4 | 25.6 | 50.1 | 60.9 | 63.6 | 43.7 | 52.6 |
| FineR Liu et al. (2024b) | BLIP-2 | Qwen-L-7B | 64.7 | 52.7 | 44.2 | 61.4 | 43.9 | 53.9 | 44.3 | 62.8 | 63.9 | 69.7 | 52.2 | 60.1 |
| Ours | BLIP-2 | Qwen-L-7B | **65.1** | **62.3** | 45.6 | 64 | 47.3 | 63.1 | 45.6 | 64 | 64.8 | 67.6 | 53.7 | 64.2 |
| FineR Liu et al. (2024b) | BLIP-2 | ChaGPT | 63.8 | 51.3 | 48.1 | 64.9 | 49.2 | 63.5 | 51.1 | 69.5 | 72.9 | 72.4 | 57.0 | 64.3 |
| Ours | BLIP-2 | ChaGPT | 64.5 | 52.1 | 50.3 | **65.6** | **50.5** | **64.7** | **51.8** | **70.6** | **73.5** | **74.2** | **58.1** | **65.4** |

## 9.3 COMPARED METHODS.

Given that our setting is a relatively new task, existing baseline models are not available in the literature. To this end, we introduce several strong baselines: (**i**) "Zero-shot (UB)" and "Finedefics (UB)" denote the upper-bound performance achieved by directly using ground-truth class names for inference and leveraging them for training, respectively. (**ii**) VQA models (BLIP-2 Flan-T5$_{\text{xxl}}$ Li et al. (2023a), MiniGPT4 Zhu et al. (2023a) and Qwen-VL Bai et al. (2023)), that identifies the object category by answering a question of the template *"What is the species of the {superclass} depicted in the provided image?"*. (**iii**) FineR Liu et al. (2024b) first extracts fine-grained visual attributes from images as text and then leverages the world knowledge of large language models (LLMs) to reason about category names. We also incorporate the prior category names into the aforementioned baselines to ensure a fair comparison with our method. (**iv**) Several category discovery approaches, such as SMILE Du et al. (2023) and PHE Zheng et al. (2024), are included as strong competitors.

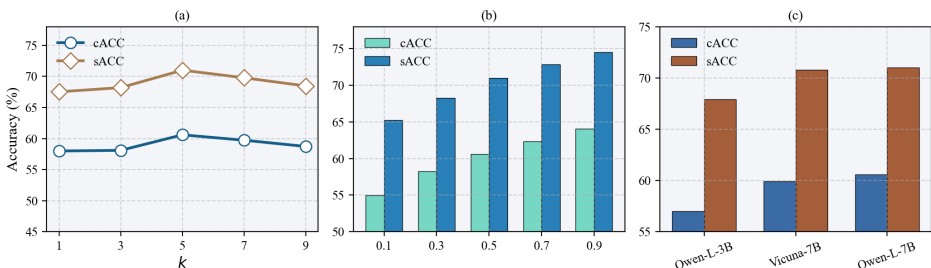

Figure 8: (a) Effect of the number of explicit anchors. (b) Effect of prior category pool with fixed number of explicit anchors. (c) The effect of LLMs capability. Experimental results are averaged over five benchmark datasets. Unless otherwise specified, BLIP-2 is used as the default VQA model and Qwen-L-7B as the default LLM.

## 9.4 ADDITIONAL COMPARISONS

**Qualitative comparison.** We present qualitative comparisons across Dog-120, Flower-102, and Pet-37 datasets in Fig. 6. Experimental results show that our SeVA framework exhibits superior fine-grained reasoning capability compared to other models, such as MiniGPT4 and BLIP2. Specifically, on the Dog-120 dataset, our method provides more fine-grained and semantically correct predictions, e.g., correctly identifying "African Hunting Dog" and "Sealyham Terrier", while other models produce coarse-grained or incorrect results. On the Flower-102 dataset, our method not only achieves the exact ground-truth label "Mexican Aster", but also surpasses the ground-truth with a more specific prediction "Pink Mexican Aster" in some case. Similar results can be observed in Pet-37 dataset. These results indicate that our method is capable of capturing subtle visual cues and leveraging them through image-specific, discriminative question generation.

**Quantitative Comparisons.** We evaluate our SeVA framework under the same setting with FineR, where only three unlabeled training images per category are available (i.e., $D_c^{\text{train}} = 3$). As shown in Tab. 4, our method consistently outperforms existing approaches across five fine-grained datasets. Compared to the SOTA method FineR Liu et al. (2024b), SeVA improves the average accuracy by +1.5% in cACC and +4.1% in sACC when both are implemented with BLIP-2 and Qwen-L-7B. When with the more powerful ChatGPT as the LLM, our approach still achieves absolute gains of +1.1% in cACC and +1.1% in sACC, demonstrating the effectiveness of our self-questioning reasoning mechanism.

In contrast, prior knowledge-based methods such as WordNet Miller (1995), SCD Han et al. (2023), and CaSED Conti et al. (2023) achieve strong performance primarily on specific domains like dogs or flowers, largely due to domain-specific coverage in their external knowledge bases. For example, WordNet and SCD fully cover all ground-truth categories in Dog-120, while CaSED's PMD knowledge base Singh et al. (2022) includes 101 out of 102 ground-truth categories in Flower-102. However, these methods exhibit limited generalization across diverse datasets. In comparison, our SeVA framework does not rely on pre-defined knowledge graphs or pseudo-labels, and can achieve superior overall performance by dynamically identifies discriminative visual cues.

**Benchmarking on Pokemon dataset.** To further assess the fine-grained visual reasoning capabilities of our framework on unfamiliar concepts, we conduct experiments on a newly curated Pokemon-10 dataset, following the setting in Liu et al. (2024b). As shown in Fig. 7, WordNet, BLIP-2, and CaSED exhibit limited performance, either due to the absence of relevant Pokémon category names in their knowledge bases or the high visual similarity among intra-class samples. FineR, a recent strong baseline, leverages the reasoning capabilities of large language models to achieve significant performance improvements on the Pokemon dataset. Different from FineR, which relies on fixed templated questions for all images, our method dynamically adapts the questioning strategy based on image content, leading to improved recognition accuracy (i.e., +0.8% cACC and +1.6% sACC).

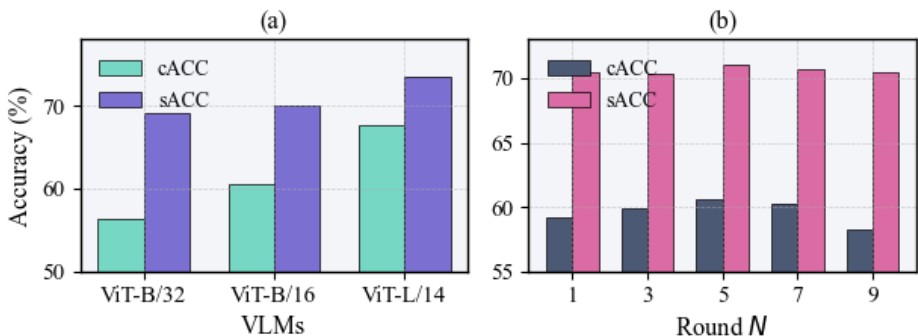

Figure 9: (a) Effect of CLIP VLMs. (b) Effect of iteration rounds. Experimental results are averaged over five benchmark datasets. Unless otherwise specified, BLIP-2 is used as the default VQA model and Qwen-L-7B as the default LLM.

## 9.5 ANALYSIS OF HYPERPARAMETERS AND MODEL CAPACITY

**Effect of the number of explicit anchors.** We investigate how the number of explicit anchors influences the average performance of our framework by varying the top-$k$ semantically similar categories used as anchors. As shown in Fig. 8 (a), the optimal performance in terms of both cACC and sACC is achieved when $k = 5$. Using too few anchors (e.g., $k = 3$) offer limited semantic cues, resulting in weaker focus and suboptimal reasoning. Conversely, too many anchors (e.g., $k = 7$) may introduce noise, distracting the model with irrelevant semantics. These results suggest that a moderate number of well-aligned explicit anchors effectively ensures high-quality initial focus, enabling the Verifier to concentrate on the most informative visual regions for precise reasoning.

**Effect of prior category pool with fixed number of explicit anchors.** To evaluate the impact of prior knowledge size, we vary the number of prior categories while keeping the top-$k$ explicit anchors fixed at $k=5$. As shown in Fig. 8 (b), the overall performance improves as the number of prior categories increases. A larger prior category pool not only reduces the number of classes that require reasoning but also provides more semantically aligned anchors. Notably, fixing $k$ avoids redundancy and allows the Verifier to consistently attend to discriminative regions guided by a compact and relevant anchor set. These results demonstrate that our method benefits from a larger prior category space with a fixed number of explicit anchors.

**The effect of LLMs capability.** In Fig. 8 (c), we investigate the effect of the large language models (LLMs) capability on the average performance of our SeVA system by comparing Qwen-L-3B and Qwen-L-7B as the Questioner and Reasoner components. As shown in Fig. 8 (c), We observe a clear positive correlation between LLM capacity and method performance. Specifically, when replacing Qwen-L-3B with the larger Qwen-L-7B model, we observe an average improvement of +5.2% in cACC and +1.6% in sACC across benchmarks. This indicates that larger LLMs are more capable of generating discriminative sub-questions and aggregating multi-step visual evidence, thereby enhancing fine-grained recognition. These results highlight the potential of further boosting SeVA's reasoning capabilities by incorporating more advanced LLMs.

We further replace the LLM of our method with Vicuna-7B Chiang et al. (2023), and compare its performance against Qwen-L-7B. As shown in Fig. 8 (c), the average performance remains stable across the two models, with only marginal differences in both cACC and sACC. This demonstrates that our method can effectively leverage different LLMs to perform fine-grained reasoning based on visual cues. These findings demonstrate the robustness and generalizability of our framework across diverse LLM backbones.

**Effect of VLMs.** We examine how the performance of our system varies with different CLIP backbones, including ViT-B/32, ViT-B/16, and ViT-L/14. As shown in Fig. 9(a), larger VLM backbones lead to consistently higher performance. Notably, when using the CLIP ViT-L/14 model, our method yields a noticeable performance gain in both cACC and sACC. This positive correlation

highlights the benefit of leveraging more powerful vision-language representations, indicating that our method can be further enhanced by scaling up the VLM backbone.

**Effect of iteration rounds.**    We study the impact of reasoning iterations $N \in \{1, 3, 5, 7, 9\}$ on recognition performance, as shown in Fig. 9(b). The results indicate that both cACC and sACC benefit from iterative reasoning, with noticeable gains from $N = 1$ to $N = 5$. This demonstrates that multi-round reasoning helps refine the model's comprehension of visual-semantic cues. However, the performance slightly declines or saturates beyond $N = 5$, suggesting that excessive iterations may introduce noise or redundant cues. Overall, our framework achieves stable performance within a small number of rounds, highlighting its efficiency and robustness.

