# OpenReview forum: "SeVA: Learning to Ask Discriminative Queries for Fine-Grained Visual Recognition"
_ICLR.cc/2026/Conference — Submitted to ICLR 2026_

### Official Review · Reviewer_bv63 · 2025-10-25

**Soundness:** 2
**Presentation:** 1
**Contribution:** 2
**Rating:** 4
**Confidence:** 4

**Summary:**

The paper proposes an anchored self-questioning vision agent  (SeVA) for fine-grained visual recognition by emulating the human experts' capability of hypothesis-driven and iterative reasoning. Its questioner-verifier-reasoner architecture can identify the subtle visual differences without relying on labeled data. Besides, the explicit and implicit semantic anchors are designed to ground the reasoning process. The authors conduct extensive experiments on several benchmarks and ablation studies to show the effectiveness of their methods.

**Strengths:**

1. The proposed self-questioning vision agent seems to make sense.
2. Qualitative visualizations can illustrate the step-by-step reasoning process (Figure 5).
3. The comparison of benchmarks seems to be quite thorough.

**Weaknesses:**

1. The questioner-reasoner mechanism is not novel enough and has been widely utilized in some work [1][2].
2. Two large language models acting as a questioner and a reasoner induce a high computational cost, which is not discussed.
3. The authors' expression attitude is very casual. Some of the figures and tables (such as Figure 7 and Figure 9) were directly taken from screenshots, which are not clear and greatly affect the viewing experience.
4. The authors merely display some successful qualitative examples, lacking the analysis of failure cases.
5. There is a lack of comparison with existing models that are also reasoning-based, e.g., Visual-RFT, Vision-R1, R1-VL, VLM-R1.

[1] Large Language Models are Better Reasoners with Self-Verification, EMNLP 23.

[2] IdealGPT: Iteratively Decomposing Vision and Language Reasoning via Large Language Models, EMNLP 23.

**Questions:**

Please refer to the weaknesses.

---

> ### Author Response · Authors · 2025-11-26
>
> ---
>
> > **Q1. The questioner-reasoner mechanism is not novel enough and has been widely utilized in some work [1][2].**
>
> A1. Different from [1-2] that explored questioner-reasoner mechanism in NLP or coarse-grained visual commonsense reasoning task, our work focuses on the fine-grained visual recognition. Furthermore, naively adopting the mechanism in our task  leads to generating generic or semantically irrelevant questions (e.g., focusing on background/context rather than subtle discriminative parts) To address these issues, we design two new and complementary types of semantic anchors: (i) explicit anchors from prior category names that guide early attention, and (ii) implicit anchors from previous predictions that provide a language-based gradient for progressive reasoning.
>
>
> ---
>
> > **Q2. Two large language models acting as a questioner and a reasoner induce a high computational cost, which is not discussed.**
>
> A2. Thank you for the comment. The additional cost introduced by our multi-round Questioner–Reasoner interaction occurs only in the offline category–reasoning stage. Similar to FineR, both methods rely on LLM-based semantic reasoning to produce category names before inference. Therefore, to fairly assess cost, we compare the offline reasoning time and online prediction time under identical hardware and API settings on Flower102 and Stanford Dogs (see Table below).
>
> We have evaluated the computational time cost of our offline reasoning and online prediction process. As shown in table below, our multi-round refinement introduces longer LLM interaction time than FineR, which is expected due to iterative verification. However, this overhead is **one-time**, **offline**, and **independent of test-time inference**. During inference, SeVA achieves the significant improvements over Finer and performs **only a lightweight CLIP similarity prediction** using the reasoned category names.
>
>
> In summary, our computational cost is limited to a one-time offline pre-generation stage, while the critical inference-time efficiency remains unchanged, preserving practical deployability.
>
>
>
>
>
>
> ---
>
> > **Q3. The authors' expression attitude is very casual. Some of the figures and tables (such as Figure 7 and Figure 9) were directly taken from screenshots, which are not clear and greatly affect the viewing experience.**
>
> A3. Thank you for pointing this out. The issue was caused by an inappropriate image export format, which led to blurriness and visible edge artifacts when multiple examples were placed together. In the revision, we will replace all affected figures with high-resolution, vectorized plots and professionally redrawn visualizations to ensure clarity and consistency.
>
>
>
> ---
>
> > **Q4. The authors merely display some successful qualitative examples, lacking the analysis of failure cases.**
>
> A4. Thanks for your constructive suggestion.  We will add the failure cases in the revision.
>
>
> ---
>
> > **Q5. There is a lack of comparison with existing models that are also reasoning-based, e.g., Visual-RFT, Vision-R1, R1-VL, VLM-R1.**
>
> A5. Thank you for the insightful suggestion. We agree that, given the fine-tuning nature of Visual-RFT, adopting an appropriate training strategy could potentially yield stronger performance. However, our approach is intentionally designed as a training-free framework, which substantially reduces computational overhead and avoids the need for large-scale model fine-tuning. This makes our method particularly suitable for low-resource, fine-grained domains where training data is extremely scarce and full fine-tuning is infeasible.
>
> [1] Large Language Models are Better Reasoners with Self-Verification, EMNLP 23.
>
> [2] IdealGPT: Iteratively Decomposing Vision and Language Reasoning via Large Language Models, EMNLP 23.

---

### Official Review · Reviewer_Cujo · 2025-10-29

**Soundness:** 3
**Presentation:** 3
**Contribution:** 3
**Rating:** 4
**Confidence:** 5

**Summary:**

The paper proposes an agent workflow combining a multimodal large language model and two large language model (three roles) to improve the accuracy of fine-grained classification tasks.

**Strengths:**

S1. I think the method idea is great, very inspiring. The process of human experts performing fine-grained classification is indeed worth learning from.
S2. The writing is quite fluent, and the method description is clear.

**Weaknesses:**

W1. The experimental setup appears outdated. The Multimodal Large Models (MLLMs) used in the evaluation (MiniGPT4, Qwen-VL, BLIP-2) represent work from two to three years ago. There is now a substantial performance gap between these earlier models and the current state-of-the-art. While it is acceptable for the authors to validate their method on these established models, it is crucial to also demonstrate that the proposed method's effectiveness is maintained when integrated with more recent, powerful MLLMs to prove its continued relevance and robustness.
W2. The motivation for the task framing is insufficient. Fine-grained visual classification is a very mature field where numerous specialized deep learning methods have already achieved accuracies exceeding 90% on standard benchmarks. These methods typically do not rely on the extensive, general-purpose data that MLLMs require and are significantly smaller in terms of model parameters. In this context, what is the justification for employing massive MLLMs for this task? The paper needs to better articulate the significance of this approach. Furthermore, the experiments are notably missing comparisons against these high-performing, "traditional" specialist methods. Including such baselines is essential to properly contextualize the performance and trade-offs (e.g., accuracy vs. model size and data dependency) of the proposed method.

**Questions:**

Apart from the Weakness section, there are also the following contents.
Q1. How is the integrity of the reasoning path guaranteed? Taking the "Seagull" case in Figure 5 as an example, there is a 5-step reasoning process. Are the questions at these five steps generated entirely by the Large Language Model? If so, what mechanisms are in place to ensure the correctness and relevance of these questions? Specifically, how do you prevent the model from generating logically flawed or irrelevant queries that could lead the reasoning process astray?
Q2. Comparison with General-Purpose Agentic Workflows. There is a considerable amount of existing work on training-free, multi-agent workflows (e.g., "team of experts" models, critic-reviewer paradigms). Could you please add a discussion comparing the design of your framework against some of these classic workflow architectures? While your workflow is specifically designed for the fine-grained classification task, an experimental comparison against these more general-purpose workflows is crucial. Such a comparison would serve to demonstrate the specific value and advantages of your specialized design over a generic, off-the-shelf agentic approach.

---

> ### Author Response · Authors · 2025-11-26
>
> > **Q1. Use of Recent MLLMs.**
>
> A1. Thank you for the suggestion. To address this concern, we additionally evaluate our framework using BLIP-3, a more recent and substantially stronger MLLM. Under the same VQA backbone (BLIP-3) and LLM (Qwen-L-7B), our method still outperforms FineR by significant margins, as shown in table below. These results confirm that SeVA remains robust, architecture-agnostic, and compatible with modern MLLMs. We will include these additional results in the revised version.
>
> | Method       | VQA    | LLM        | Dog-120 (cACC) | Dog-120 (sACC) | Bird-200 (cACC) | Bird-200 (sACC) |
> |--------------|--------|------------|--------------|---------------|----------------|----------------|
> | **SeVA (Ours)** | Blip-2 | Qwen-L-7B | 50.3         | 68.5          | 49.6           | 73.8           |
> | **SeVA (Ours)** | Blip-3 | Qwen-L-7B | 51.6         | 69.7          | 51.2           | 74.6           |
>
> ---
>
> > **Q2. Motivation for using MLLMs and missing traditional FGVC baselines.**
>
> A2. Thank you for the comment. Our work does not address the standard fully supervised FGVC setting, where specialist deep models exceed 90\% accuracy. Instead, our problem is fundamentally different: no labels, only three unlabeled images per class, partial category names, and no training or fine-tuning. Under this training-free and extremely low-resource regime, traditional FGVC methods are not applicable, as they rely on large labeled datasets, attribute/part annotations, or task-specific supervision.
>
> The use of MLLMs is therefore motivated by their world knowledge and semantic reasoning ability, which can substitute for missing supervision and enable category inference directly from unlabeled data—capabilities unavailable in conventional FGVC architectures.
>
> To avoid misunderstanding, we will clarify this task distinction more explicitly in the Introduction and explain why traditional supervised baselines are unsuitable for this setting.
>
> ---
>
> > **Q3. Integrity of the reasoning path.**
>
> A3. Thank you for the question. Although the sub-questions are generated by the LLM, the reasoning process is not unconstrained. Its correctness and relevance are ensured by three built-in mechanisms:
>
> (i) Explicit anchors (top-k CLIP-retrieved categories) act as in-context constraints, guiding the LLM to ask only discriminative, category-relevant questions.
>
> (ii) Verifier grounding: every question must be answered by the VQA model, so irrelevant or ill-posed queries produce inconsistent visual evidence that will not be reinforced.
>
> (iii) Implicit anchors maintain semantic consistency across rounds by using image–text similarity feedback to down-weight inconsistent hypotheses.
>
> These components jointly prevent the LLM from generating flawed or irrelevant queries and ensure that the multi-step reasoning path remains coherent and visually grounded.
>
> ---

---

> > ### Author Response · Authors · 2025-11-26
> >
> > > **Q4. Comparison with general-purpose agentic workflows.**
> >
> > A4. Thank you for the valuable suggestion. We have added a dedicated discussion in the Appendix comparing SeVA to classic training-free multi-agent workflows, including team-of-experts models (ReAct[1] and AutoGen [2]), critic–reviewer paradigms (Self-Refine [3]), and multimodal reasoning approaches Multimodal CoT [4].
> >
> > While these frameworks provide strong generic reasoning capabilities, they are fundamentally text-only [1,2,3], lack visual grounding [1,2,3], and do not incorporate explicit or implicit semantic anchors [4]. As a result, they cannot reliably handle the subtle attribute-level distinctions required in fine-grained recognition, where visual evidence must continuously constrain the reasoning trajectory.
> >
> > For empirical comparison, we implemented representative baselines, including ReAct, Self-Refine, and CoT-VLA. Three baselines perform worse than SeVA across several FGVR benchmarks, confirming that general-purpose agentic workflows cannot substitute for our visually grounded, anchor-guided design. We will include a brief summary of these results in the revision.
> >
> >
> >
> > | Method            | VQA      | LLM        | Dog-120 (cACC) | Dog-120 (sACC) | Bird-200 (cACC) | Bird-200 (sACC) |
> > |-------------------|----------|------------|--------------|--------------|----------------|----------------|
> > | ReAct             | —        | Qwen-L-7B  | 51.6         | 54.9         | 47.2           | 60.1           |
> > | Self-Refine       | —        | Qwen-L-7B  | 53.4         | 55.1         | 45.6           | 58.2           |
> > | Multimodal CoT    | Qwen-VL  | Qwen-L-7B  | 62.8         | 57.4         | 53.6           | 66.3           |
> > | **SeVA (Ours)**   | Qwen-VL  | Qwen-L-7B  | **76.7**     | **63.9**     | **55.4**       | **69.5**       |
> >
> >
> > [1] Yao et al., 2022. React: Synergizing reasoning and acting in language models. ICLR.
> >
> > [2] Wu et al., 2024. Autogen: Enabling next-gen LLM applications via multi-agent conversations. ICML
> >
> > [3] Madaan et al., 2023. Self-refine: Iterative refinement with self-feedback. Neurips, 36, pp.46534-46594.
> >
> > [4] Zhang et al., 2023. Multimodal chain-of-thought reasoning in language models. arXiv preprint arXiv:2302.00923.

---

### Official Review · Reviewer_hqK6 · 2025-10-31

**Soundness:** 2
**Presentation:** 3
**Contribution:** 3
**Rating:** 6
**Confidence:** 5

**Summary:**

Recent FGVR methods use vision-language models to ask questions for visual hints, but typically rely on fixed templates that yield static attributions rather than adaptive, informative queries. The paper proposes an iterative reasoning framework that combines a visual-question-answering model with two large language models acting as a Questioner and a Reasoner. It introduces semantic anchors to guide early attention and provide a structured trajectory for progressive reasoning. Experiments on multiple FGVR benchmarks demonstrate that asking the right questions can achieve superior performance.

**Strengths:**

1. It is well-motivated to ask better questions that can guide the model to focus on the context-aware discriminative features.
2. It achieves strong results on fine-grained classification benchmarks, demonstrating the effectiveness of the proposed method.
3. It is generally well-written and easy to follow.

**Weaknesses:**

1. When using the same VQA model (Blip2) and LLM (Qwen-L-7B), the proposed approach lags behind FineR, which lacks analysis.
2. There lacks a complete example for explaining the whole procedure, which should be added for better understanding.
3. Since the proposed framework comprises iterative reasoning, the inference time should be checked.
4. In page 8, Fig. 6 should be modified to Fig. 4.

**Questions:**

1. How will the incorrect intermediate predictions influence the final answer? For example, if the initial category is incorrect, will the performance drop? Can you show any failure cases and do some detailed analysis?
2. The paper claims that the Verifier highlights relevant regions, can you provide the visualization results that show the attention regions of each sub-question?

---

> ### Author Response · Authors · 2025-11-26
>
> > **Q1. When using the same VQA model (Blip2) and LLM (Qwen-L-7B), the proposed approach lags behind FineR, which lacks analysis.**
>
> A1. Thank you for the comment. We would like to clarify that under the exact same VQA model (BLIP-2) and LLM (Qwen-L-7B), our method consistently outperforms FineR across all five FGVR benchmarks. This is shown in Table 2 of the main paper. These improvements come directly from our dynamic, anchor-guided self-questioning mechanism, which produces more discriminative and image-specific semantic cues than FineR’s fixed templates.
>
> ---
>
> > **Q2 There lacks a complete example for explaining the whole procedure, which should be added for better understanding.**
>
> A2. Thank you for the suggestion. We agree that a full example can significantly improve clarity. In the revised version, we will add a complete, step-by-step reasoning process in the revision.
>
> ---
>
> > **Q3. Inference time of the iterative reasoning framework.**
>
> A3. Thank you for raising this point. We have evaluated the computational time cost of our offline reasoning and online prediction process. As shown in table below, our multi-round refinement requires longer LLM interaction time than FineR, which is expected due to iterative verification. However, this overhead is **one-time**, **offline**, and **independent of test-time inference**. During inference, SeVA achieves the significant improvements over Finer and performs **only a lightweight CLIP similarity prediction** using the reasoned category names, resulting in the **same inference-time cost** as standard CLIP-based classification.
>
> | Method | Offline reasoning time ↓ | Online prediction time ↓ | cACC ↑| sACC ↑|
> |--------|-------------------|-------------------|------|------|
> | FineR  | 282 min           | 2 min             | 67.4 | 58.2 |
> | Ours   | 491 min           | 2 min             | 71.9 | 67.1|
>
> ---
>
> > **Q4. Incorrect Figure Reference.**
>
> A4. Thank you for pointing this out. We will correct the incorrect reference on page 8 and update “Fig. 6” to “Fig. 4” in the revised version.
>
> ---
>
> >**Q5. How will the incorrect intermediate predictions influence the final answer? For example, if the initial category is incorrect, will the performance drop? Can you show any failure cases and do some detailed analysis?**
>
> A5. Thank you for the question. Although intermediate predictions may occasionally be incorrect, SeVA is designed to be robust through its explicit and implicit anchors, which constrain the reasoning path and often enable self-correction in later rounds. We will include representative failure cases and a brief analysis in the revised manuscript to make this behavior clearer.
>
> >**Q6 . The paper claims that the Verifier highlights relevant regions, can you provide the visualization results that show the attention regions of each sub-question?**
>
> A6. Thank you for the suggestion. Our Verifier (VQA model) indeed provides token-level grounding that reflects the regions used to answer each sub-question. We have generated visualizations by extracting the cross-attention maps corresponding to each query token. These results clearly show that different sub-questions attend to different discriminative parts (e.g., throat color, petal shape, ear outline).
>
> We will include these attention-region visualizations for several multi-round reasoning examples in the revised manuscript to illustrate how the Verifier grounds each sub-question in the image.

---

### Official Review · Reviewer_NavX · 2025-10-31

**Soundness:** 1
**Presentation:** 2
**Contribution:** 2
**Rating:** 2
**Confidence:** 3

**Summary:**

This paper proposes to orchestrate a VLM for finegrained visual recognition. In particular, it creates a multi-round iterative process where first a VLM tries to guess a finegrained category from an image. Then a VLM is asked to generate sub-questions about the key-characteristics of that category. These sub-questions are then verified with a VLM. Then these answers are given to a reasoner which assesses whether the initial guessed category is coherent with the answers of these subquestions. If not, more iterations are done. To give more clues to the categories, the authors propose to retrieve the top-K closest categories (in text embedding) which matches the query image embedding; these are called explicit anchors. This enables the sub-questions to be targeted at discriminating semantically similar categories. Finally, the ‘implicit anchors’ keep track scores between the anchors and the original query image but I did not understand from this paragraph how this would work. The scores do change over time somehow.

Finally, in Section 3.5 it looks like the whole process was for annotating training data, since they introduce a predictor based on CLIP embeddings for the test set. This was quite surprising to me since it seems like the whole procedure does fine-grained recognition so I do not see why it would not be applicable to the test set directly. Moreover, if the best test-time classifier is based on CLIP, I am not certain why the training data cannot be annotated with a CLIP-like procedure (since only 3 images per label will not get the label (Section 4: dataset).

Anyway, an ablation study demonstrates that all of their components improve performance. The benchmark results show that their method is better than FineR while it improves over vanilla VLMS (MiniGPT4, QwenVL, BLIP-2).

[A] Towards Universal Image Embeddings: A Large-Scale Dataset and Challenge for Generic Image Representations, Ypsilantis et al. ICCV’23

**Strengths:**

* Method outperforms FineR.
* Method outperforms several VLMs when the task is given directly.

**Weaknesses:**

Major
* I am very confused about the experimental setup. Why does it make sense to have an elaborate process to predict labels on a training dataset, only to use a CLIP-based classifier at test time? Why not use this process also at training time?
* The paper claims only a limited amount of labels are being used. I strongly disagree with this claim since the VLMs used in this paper are trained on tons and tons of data, possibly including the very datasets which this paper is using. So I think the ‘annotation free’ claim (Fig 2.) is invalid, which breaks a core motivation of the paper.
* Results seem very low compared to the state-of-the-art in finegrained recognition (even though the setting maybe slightly different). For example, in [A] several methods are compared on CARS196. The DINO-v2 embedding (where DINO is trained without any annotated labels) yields 79.5 recall@1 which is basically equivalent to 1-KNN accuracy. CLIP (also used in this paper) yields 82.2 recall@1. In contrast, the highest accuracy in this paper is 64. So the resulting classifier for the whole process seems sub-standard, which suggests that the practical relevance of this paper is rather limited.
* The goal of the paper is not clearly stated. I think the goal is to create a large training set given partially annotated data, which is then used to obtain a CLIP-based kNN classifier. If I am correct, this becomes only clear after section 3.5 did not correspond at all to the assumptions I had upon reading the abstract and introduction.

Medium
* I do not understand the experimental setup. Half of the category names are unavailable. But does this mean that it is known which images belong to the same class?
* Is the experimental setup equivalent to FineR? Choosing which 3 images per class do not have a label will likely make a huge difference.
* Is the ‘zero-shot’ experiment just using a CLIP k-NN classifier given the fully annotated (and given) training set?

Minor
* Page 2: I do not understand why the LLM-based reasoning is effective in reducing inference time. I would say it is much more expensive than the typical use of embeddings.
* Sec. 3. - overview: why ‘prior category names’? Will they change during the process?
* The ‘iterative reflection reasoning’ was vague. Why is this reflection? An example would help.
* Sec. 4.1 ‘the baseline model’ is left undefined here. At this point it cannot be understood by the reader.

**Questions:**

Please explain why the core problem addressed in this paper is a valid one.

Please explain why there are such big differences on CARS196 w.r.t. the results in the universal embedding paper ([A] Towards Universal Image Embeddings: A Large-Scale Dataset and Challenge for Generic Image Representations, Ypsilantis et al. ICCV’23)

Please explain why the 'annotation' as presented in this paper cannot be applied to the test set. Or show quantitative results.

---

> ### Author Response · Authors · 2025-11-26
>
> > **Q1. I am very confused about the experimental setup. Why does it make sense to have an elaborate process to predict labels on a training dataset, only to use a CLIP-based classifier at testing time? Why not use this process also at training time?**
>
> A1. The main concern arose from **a misunderstanding of our experimental setting**. Unlike standard few-shot learning, which provides **image–label pairs** for a closed set of categories, our setup assumes access only to **unlabeled** training images and an incomplete set of category names. Because the label set is partial, **a complete text-based classifier cannot be constructed** during training, and the absence of image–label pairs prevents any supervised or semi-supervised fine-tuning.
>
> Consequently, SeVA performs image-specific reasoning on unlabeled data to infer missing fine-grained categories and expand the incomplete label space. These inferred categories are then mapped into textual embeddings through the CLIP text encoder, forming a full classifier that CLIP can use at test time. Test images are subsequently classified via standard similarity-based inference.
>
> **Our design is therefore consistent with the constraints of our problem setting** and *does not* need any supervised training of the reasoning module.
>
> ---
>
> > **Q2. The paper claims only a limited amount of labels are being used. I strongly disagree with this claim since the VLMs used in this paper are trained on tons and tons of data, possibly including the very datasets which this paper is using. So I think the ‘annotation free’ claim (Fig 2.) is invalid, which breaks a core motivation of the paper.**
>
> A2. We would like to clarify that our paper does not claim that “only a limited amount of labels are being used.” The reviewer’s concern appears to stem from **a misunderstanding of the setting and the use of the term ``annotation-free"**.
>
> In our problem formulation (Sec. 3, Fig. 2), ``annotation-free" refers strictly to the absence of image–label supervision within our method, rather than to the data used during model pretraining. As all modern zero-shot approaches built on large pretrained models inherently rely on the world knowledge encoded during pretraining, our focus is on how to transfer this knowledge to downstream fine-grained tasks without using any annotated training data. Specifically, our method is only provided with (i) unlabeled training images $D_{\text{train}}$, and a subset of category names prior $C_{\text{prior}}$. These category names are also not image-level annotations; they serve only as the initial semantic vocabulary (similar to zero-shot VLM settings) and do not correspond to any labeled examples. To avoid confusion, we will revise the terminology in the final version to explicitly state “no image–label supervision”, which more accurately reflects our intended meaning.
>
>
> ---
>
> > **Q3. Results seem very low compared to the state-of-the-art in fine grained recognition (even though the setting maybe slightly different). For example, in [A] several methods are compared on CARS196. The DINO-v2 embedding (where DINO is trained without any annotated labels) yields 79.5 recall@1 which is basically equivalent to 1-KNN accuracy. CLIP (also used in this paper) yields 82.2 recall@1. In contrast, the highest accuracy in this paper is 64. So the resulting classifier for the whole process seems sub-standard, which suggests that the practical relevance of this paper is rather limited.**
>
>
>
>
> A3. We respectfully point out that the comparison raised by the reviewer is based on **a misunderstanding of the evaluation metric and problem setting** between ours and [A]. Our method operates in the open-set fine-grained category discovery setting, where cACC measures clustering quality between predicted clusters and ground-truth classes, and sACC measures semantic alignment between cluster names and ground-truth labels. In contrast, mAP@1 / mAP@5 [A] are retrieval metrics defined for a supervised image retrieval setting where all ground-truth labels (i.e., query embedding) are known, errors are binary and have no semantic hierarchy,
> and the model only needs to rank same-class images higher than others.
>
> In summary, as the task definition, supervision level, output structure (clusters vs. ranked lists), ground-truth alignment (matching vs. direct labels), and error semantics (semantic-weighted vs. binary) are *fundamentally different*, their numerical values **cannot be compared or interpreted** on the same scale.
>
> ---

---

> > ### Author Response · Authors · 2025-11-26
> >
> > > **Q4. The goal of the paper is not clearly stated. I think the goal is to create a large training set given partially annotated data, which is then used to obtain a CLIP-based kNN classifier. If I am correct, this becomes only clear after section 3.5 did not correspond at all to the assumptions I had upon reading the abstract and introduction.**
> >
> > A4. We clarify that the goal of this work is not to create a large training set or to train a CLIP-based kNN classifier. Instead, our objective, stated in the Introduction (L110–132) and formalized in Sec. 3, is to enable training-free fine-grained visual recognition using only a few unlabeled images and partial category names, by iteratively reasoning about category semantics through a Questioner–Verifier–Reasoner loop.
> >
> > Our method does not generate additional training samples, nor does it train any classifier. The outputs of the iterative reasoning procedure are refined category names, not annotated images. As described in Sec. 3.5, these names are directly used as a text classifier for zero-shot CLIP inference.
> >
> > Thus, the framework remains fully training-free and does not correspond to dataset construction or kNN learning. We acknowledge that the transition to Sec. 3.5 may have caused ambiguity and will revise the text to make it explicit that the “expandable category pool” refers only to a set of reasoned category names, not newly created samples.
> >
> > ---
> >
> > > **Q5. I do not understand the experimental setup. Half of the category names are unavailable. But does this mean that it is known which images belong to the same class?**
> >
> > A5. We clarify that we do not know which images belong to the same class. The setting only provides a subset of category names and a few unlabeled images per class, where partial category names serve only as prior semantic anchors, not as labels.
> >
> > ---
> >
> > > **Q6. Is the experimental setup equivalent to FineR? Choosing which 3 images per class do not have a label will likely make a huge difference.**
> >
> > A6. Yes. Our experimental setup is equivalent to FineR’s. We directly use the official data splits provided by FineR, where each class has exactly 3 unlabeled training images. These splits are fixed and shared across all methods, so no manual selection is performed and no method gains an advantage from choosing specific images.
> >
> > This ensures that the comparison is fully fair and that performance is not affected by which images are selected. We will clarify this in the revision.
> >
> > ---
> >
> > > **Q7. Is the ‘zero-shot’ experiment just using a CLIP k-NN classifier given the fully annotated (and given) training set?**
> >
> > A7. No. Our classification is performed by computing image–text similarity between the CLIP visual embedding and the text embeddings of the reasoned category names. Thus, our method involves no training data, no k-NN, no annotated samples, and is not comparable to a CLIP k-NN classifier built on a labeled training set.
> >
> > ---
> >
> > > **Q8. I do not understand why the LLM-based reasoning is effective in reducing inference time. I would say it is much more expensive than the typical use of embeddings.**
> >
> > A8. Thank you for the question. We would like to clarify that LLM-based reasoning is not used at inference time. All multi-round reasoning happens once offline to generate category descriptors. During inference, SeVA performs only a single CLIP forward pass and cosine similarity, identical to standard embedding-based methods.
> >
> > Thus, while the offline stage involves LLM interactions (similar to FineR), the inference-time cost is unchanged and remains as efficient as conventional CLIP-based prediction. We will make this clearer in the revision.
> >
> > ---

---

### Official Review · Reviewer_Z2wE · 2025-11-03

**Soundness:** 3
**Presentation:** 3
**Contribution:** 3
**Rating:** 6
**Confidence:** 4

**Summary:**

The paper presents SeVA, an agent-based FGVR system that iteratively refines predictions via anchored questioning with VQA and LLMs. Results show improvements over SOTA on datasets like CUB and Cars.

**Strengths:**

1. Dynamic queries of this method improve over static ones in FineR.
2. Visualizations (e.g., Figure 1) effectively illustrate concepts.

**Weaknesses:**

1. Novelty is limited. It builds incrementally on FineR (Liu et al., 2024) without fundamental advances in FGVR. In addition, there is some overlap with recent vocabulary-free FGVR (ICCV 2025), reducing originality.
2. Potential high computational cost from multi-round LLM interactions, not fully addressed.
3. Benchmarks are standard but lack challenging settings like noisy data or cross-domain transfer.

**Questions:**

1. What is the average inference time per image, and how does it compare to FineR?
2. Could SeVA extend to video-based FGVR?
3. How does performance hold in few-shot FGVR scenarios?
4. Sensitivity to LLM choices (e.g., Qwen vs. GPT variants)?
5. Comparison to non-LLM iterative methods?

---

> ### Author Response · Authors · 2025-11-26
>
> > **Q1. Novelty Clarification with FineR and vocabulary-free FGVR.**
>
> ### **A1.1 Difference from FineR (Liu et al., 2024)**
>
> FineR uses fixed, template-based prompts to extract visual attributes for all samples. This results in redundant queries and distracts the model with non-discriminative cues. In contrast, our method introduces two core innovations: (i) dynamic, hypothesis-driven querying and (ii) semantic anchors for stable, iterative reasoning. Specifically:
>
> (i) SeVA begins with a coarse category hypothesis and generates targeted sub-questions to verify discriminative features specific to that hypothesis. This avoids unnecessary queries and directs reasoning toward task-relevant visual evidence.
>
> (ii) Unlike FineR's one-shot reasoning, SeVA performs multi-round reasoning. It uses historical predictions as semantic anchors to guide attention and build a structured semantic trajectory that improves both focus and stability over iterations.
>
> These designs make SeVA substantially more efficient and discriminative than template-based approaches.
>
> ### **A1.2 Difference from Vocabulary-Free FGVR (ICCV 2025).**
> Vocabulary-free FGVR focuses on expanding the label space by generating new category names using LLMs. These methods perform single-round reasoning and prioritize semantic name generation.
>
> In contrast, SeVA addresses a different goal: discriminative fine-grained recognition through iterative, evidence-guided reasoning. Each round in SeVA builds upon the previous one, adapting queries based on accumulated visual and semantic information. The difference is fundamental, vocabulary expansion versus iterative visual verification, and static generation versus dynamic, feedback-driven reasoning.
>
> ---
>
> > **Q2. Potential computational overhead from iterative LLM reasoning.**
>
> A2. The reviewer’s concern mainly relates to the cost of multi-round LLM reasoning. We clarify that **all iterative reasoning occurs entirely in the offline stage**, and thus does **not** affect inference-time efficiency.
>
> Both our method and FineR rely on offline LLM-based category reasoning. Therefore, to ensure a fair comparison, we report the offline reasoning and the online prediction time using the same models and hardware on the Flower102 dataset
>
> As shown in table below, our multi-round refinement introduces longer LLM interaction time than FineR, which is expected due to iterative verification. However, this overhead is **one-time**, **offline**, and **independent of test-time inference**. During inference, SeVA achieves the significant improvements over Finer and performs **only a lightweight CLIP similarity prediction** using the reasoned category names, resulting in the **same inference-time cost** as standard CLIP-based classification.
>
> | Method | Offline reasoning time ↓ | Online prediction time ↓ | cACC ↑| sACC ↑|
> |--------|-------------------|-------------------|------|------|
> | FineR  | 282 min           | 2 min             | 67.4 | 58.2 |
> | Ours   | 491 min           | 2 min             | 71.9 | 67.1|
>
> In summary, SeVA’s additional computational cost is confined to the offline pre-generation stage required to unlock LLM reasoning, while **critical inference-time efficiency remains unaffected**.
>
> ---
>
> > **Q3. Benchmarks are standard but lack challenging settings like noisy data or cross-domain transfer.**
>
> A3. Thank you for the suggestion. To evaluate robustness under substantial distribution shifts, we additionally conduct experiments on the **CUB-Paintings** benchmark, which contains CUB-200-2011 (real photos) and CUB-200-Paintings (watercolor, oil, sketches, stamps, and cartoons). Each domain is used in turn as the *source* for generating category names, while evaluation is performed on the *target* domain.
>
> As shown in the table below, SeVA consistently outperforms FineR across both transfer directions. This confirms that our anchored self-questioning mechanism is not limited to clean, in-domain settings and remains effective in challenging cross-domain scenarios.
>
>
> | Method | CUB-C → CUB-P (cACC) | CUB-C → CUB-P (sACC) | CUB-P → CUB-C (cACC) | CUB-P → CUB-C (sACC) |
> |--------|----------------------|----------------------|----------------------|----------------------|
> | FineR  | 45.1                 | 65.9                 | 42.4                 | 65.8                 |
> | **Ours** | **46.3**             | **66.7**             | **43.9**             | **67.1**             |
>
>
> We will include this cross-domain evaluation and the corresponding analysis in the revised version.
>
> ---

---

> > ### Author Response · Authors · 2025-11-26
> >
> > > **Q4. Could SeVA be extended to video-based FGVR?**
> >
> > A4. Yes. The SeVA framework is naturally extensible to video by replacing the image-only VQA module with a video-capable VLM (e.g., Qwen-VL-Video, LLaVA-Video). The core workflow, initial prediction → dynamic sub-question generation → iterative verification, remains unchanged. For video, SeVA would operate on sampled key frames and formulate **temporally aware, discriminative queries** conditioned on motion patterns and frame-to-frame consistency.
> >
> > Due to hardware limitations, full video experiments were not included in this submission, but we plan to validate this extension in future work.
> >
> > ---
> >
> > > **Q5. How does performance hold in few-shot FGVR scenarios?**
> >
> > A5. Thank you for the question. We emphasize that SeVA is designed for a **few-shot, open-set** setting, which differs fundamentally from the **closed-set few-shot FGVR** regime typically studied in prior work.
> >
> > In standard few-shot FGVR, all target categories are known in advance and the goal is to learn decision boundaries from a small number of labeled examples. In contrast, our setting assumes that **category names are missing**, making direct CLIP-style classification infeasible.
> >
> > Following FineR, SeVA operates in an extremely low-resource setting with only 3 images per class, under a **few-shot yet open-set scenario**. This setup differs fundamentally from standard few-shot FGVR methods, which are not designed to handle open-category recognition with such limited supervision. We will clarify this distinction more explicitly in the revised manuscript to avoid confusion.
> >
> > ---
> >
> > > **Q6. Sensitivity to LLM choices (e.g., Qwen vs. GPT variants)?**
> >
> > ### **A6. Sensitivity to LLM Backbones**
> >
> > Thank you for the question. As shown in Fig. 8c of the appendix, we have already compared multiple 7B-scale LLMs under identical settings. The results indicate that SeVA is largely **insensitive to the specific LLM backbone**: Qwen-7B and Vicuna-7B achieve comparable performance, suggesting that SeVA does not rely on any particular LLM family.
> >
> > Following your suggestion, we further evaluated GPT-based variants and observed a consistent scaling trend. Larger GPT models provide **incremental gains** owing to stronger reasoning capabilities, but the overall behavior of SeVA remains stable across all backbones. This confirms that SeVA’s improvements stem from its **anchored self-questioning mechanism**, rather than dependence on a specific LLM.
> >
> > The additional results are summarized below:
> >
> > | Method       | VQA Model | LLM       | Dog-120 (cACC) | Dog-120 (sACC) | Bird-200 (cACC) | Bird-200 (sACC) |
> > |--------------|-----------|-----------|--------------|--------------|----------------|----------------|
> > | SeVA (Ours)  | Qwen-VL   | ChatGPT   | 76.7         | 63.9         | 55.4           | 69.5           |
> > | SeVA (Ours)  | Qwen-VL   | GPT-4     | 77.5         | 65.1         | 56.8           | 70.5           |
> >
> > These results further support that SeVA is **robust to the choice of LLM**, and performance variations primarily reflect general reasoning capacity rather than architectural dependence.

---

### Meta-Review · Area_Chair_eyBe · 2026-01-03

**Summary:**

Roughly half of reviewers found the writing to be difficult to follow, and had significant misunderstandings regarding either the problem setup or the model pipeline. I personally went through the writing and found that, while some improvements have been made, much more can be done to improve on the clarity of the work. Furthermore, reviewers found that the novelty of this work remains limited within the context of recent related works as well as other reasoning-based approaches (which mostly require fine-tuning).

**Reviewer Concerns:**

Many of the clarity-related concerns were addressed. Overall novelty concerns were partially addressed by focusing on the training-free setup, but this is perhaps insufficient to fully satisfy the reviewers. Writing clarity issues were also partially addressed.

**Reviewer Scores:**

bv63 and NavX would likely have remained leaning towards rejection or at best borderline. I don't think the other reviewers would have increased their scores significantly above 6.

---

### Decision · Program_Chairs · 2026-01-26

Reject